# The Fate of Endemic Species Specialized in Island Habitat under Climate Change in a Mediterranean High Mountain

**DOI:** 10.3390/plants11233193

**Published:** 2022-11-22

**Authors:** Antonio J. Mendoza-Fernández, Ángel Fernández-Ceular, Domingo Alcaraz-Segura, Miguel Ballesteros, Julio Peñas

**Affiliations:** 1Department of Biology and Geology, CEIMAR, CecoUAL, University of Almería, 04120 Almería, Spain; 2Department of Botany, University of Granada, 18071 Granada, Spain; 3Andalusian Center for the Assessment and Monitoring of Global Change (CAESCG), University of Almería, 04120 Almería, Spain; 4iEcolab, Inter-University Institute for Earth System Research, University of Granada, 18006 Granada, Spain; 5Department of Botany, University of South Bohemia, CZ-37005 České Budějovice, Czech Republic

**Keywords:** diversity loss, fine-scale ecological niche modeling, global change, *Moehringia fontqueri*, mountain cliff escarpments, reproductive success, Sierra Nevada (Spain)

## Abstract

Mediterranean high-mountain endemic species are particularly vulnerable to climatic changes in temperature, precipitation and snow-cover dynamics. Sierra Nevada (Spain) is a biodiversity hotspot in the western Mediterranean, with an enormous plant species richness and endemicity. *Moehringia fontqueri* is a threatened endemic plant restricted to north-facing siliceous rocks along a few ridges of the eastern Sierra Nevada. To guide conservation actions against climate change effects, here we propose the simultaneous assessment of the current reproductive success and the possible species’ range changes between current and future climatic conditions, assessing separately different subpopulations by altitude. Reproductive success was tested through the seed-set data analysis. The species’ current habitat suitability was modeled in Maxent using species occurrences, topographic, satellite and climatic variables. Future habitat suitability was carried out for two climatic scenarios (RCP 2.6 and 8.5). The results showed the lowest reproductive success at the lowest altitudes, and vice versa at the highest altitudes. Habitat suitability decreased by 80% from current conditions to the worst-case scenario (RCP 8.5). The lowest subpopulations were identified as the most vulnerable to climate change effects while the highest ones were the nearest to future suitable habitats. Our simultaneous assessment of reproductive success and habitat suitability aims to serve as a model to guide conservation, management and climate change mitigation strategies through adaptive management to safeguard the persistence of the maximum genetic pool of Mediterranean high-mountain plants threatened by climate change.

## 1. Introduction

High mountains account for 15% of the World’s temperate-zone surface [1], harboring one-third of terrestrial species [2], and more than three times the number of plants estimated by their extent [3]. Furthermore, they constitute half of the 34 global macro-hotspots [4,5], due to the high rate of richness and endemicity resulting from the biogeographic isolation of these ecosystems [6]. However, alpine biomes are particularly sensitive to changes in climatic conditions [7], both in the altitudinal gradient and over small geographic distances, since they are characterized by very strong contrasts in the living conditions for organisms [8,9].

Ongoing climate change has been reported to affect the phenology and physiology of organisms [8], the range of species [10,11], the interactions within communities and the structure and dynamics of ecosystems [12]. Mountain ecosystems, particularly in Mediterranean-climate areas, are sensitive to changes in environmental conditions across geographical and altitudinal gradients [9,13,14,15,16]. The effects of global warming can cause ecological imbalances affecting individual species and communities, most notably in the form of temperature-driven range shifts [12,17,18,19]. Mediterranean high mountains are experiencing longer summer droughts [20] and thermophilization [21,22] which can promote the spread of thermophilic species up to high mountains [10,23,24]. However, a critical aspect of biodiversity conservation is to understand how species with narrow ecological requirements (e.g., habitat specialists) and restricted distribution ranges (e.g., endemics) are altered by changes in environmental conditions.

One of the major focuses in conservation must be endemic and threatened species from regions with high biodiversity, a high degree of endemism and sensitivity to environmental changes [25]. Factors such as the small number of occupied habitat patches, restricted area of distribution and limitations of suitable habitat around their current populations [26] may determine a higher sensitivity to a threat factor. In this sense, taxa restricted to ‘high-altitude islands’ are less likely to survive a stochastic event, despite the habitat heterogeneity may provide micro-refuge [27,28]. Especially endemic species could be pushed to the brink of extinction.

Addressing uncertainty in the adaptative behavior of high mountain plants requires fundamental data on demography to assess population dynamics [29], information on genetic variability as an estimate of the ability to cope with change [30,31] and forecasting possible alterations in their distribution patterns [26,32,33,34]. For the latter assumption, species distribution models (SDMs) are useful tools to ascertain the potential distribution possibilities and allow to set the strategic conservation targets [35,36] in a climate change context [37,38]. However, knowing the potential shifts in habitat suitability should also be accompanied by an assessment of the actual reproduction potential and possible colonization of new areas of the species, particularly of island-habitat endemics. Hence, attention should be paid to reproductive biology, conservation genetics and the threats and direct effects of climate change on species distribution range [31,35,39].

Sierra Nevada is an isolated Mediterranean high massif that takes part of a heterogeneous group of Mediterranean high mountains [40]. It comprises 2348 vascular plant taxa, where 362 taxa inhabit the alpine area, 75 endemic species (62 endemic plus 13 sub-endemic) among them, constituting ca. 79% of the endemism of the entire area [41]. For this reason, Sierra Nevada is considered one of the most important plant hotspots within the Mediterranean region [42,43,44]. Restricted to the alpine belt of Sierra Nevada the species *Moehringia fontqueri* Pau (=*Arenaria funiculata* Fior & P.O. Karis) (*Caryophyllaceae*) is found (Figure 1) [45]. It is a threatened narrow endemic inhabiting crevices in siliceous shady cliffs along ridges of eastern Sierra Nevada (Almeria). It is categorized as Endangered (EN) in the Spanish National Red List [46] and is protected at regional, national and international levels. Moreover, habitat 8220* ‘Siliceous rocky slopes with chasmophytic vegetation’ is included in the European Habitat Directive as a priority habitat for conservation (DIR 92/43/EEC). Its strict ecological requirements, high habitat specificity, and its vulnerability due to climate change constitute important risk factors [47]. The preservation of *M. fontqueri* is an example of the efforts needed for designing appropriate conservation strategies for species in Mediterranean mountain areas facing climate change.

In this study, we aimed to (1) assess the variability in reproductive success of *M. fontqueri* along the entire altitudinal range and its reproductive trends over time; (2) quantify the environmental variables that explain its narrow ecological habitat requirements and distribution range, and forecast trends in habitat suitability considering future climate change scenarios; and (3) integrate the mentioned points to assess risks and improve *M. fontqueri* conservation as model species of Mediterranean high mountain endemics.

## 2. Results

### 2.1. Reproductive Performance

The production of seed primordia and viable seeds was conditioned by altitude and year, with different reproductive responses among subpopulations depending on the year, as revealed by a significant interaction between year and site in our analysis (Table 1 and Figure 2). Nevertheless, both reproductive variables exhibited an increasing homogenization over time at the three sites. In general, the species performed better (i.e., more seed primordia per flower and seeds per fruit) at the upper sites S2 and S3, which exhibited a more similar and consistent reproductive response over time. The number of seed primordia per flower and number of viable seeds per fruit reached maximum values and minimum aborts per fruit at S3 (Figure 2). In turn, S1 exhibited a worse performance for all reproductive variables and larger variation over time, suggesting a stronger dependence on the yearly climatic conditions. The number of aborts differed among the three sites and four years, without interaction (Table 1), and reached its maximum value at lower altitudes (Figure 2).

The species, accounting for the variation among sites, exhibited a slight decrease in the number of primordia over time (−0.31 ± 0.1, *p* = 0.04) as well as in the number of aborts (−0.72 ± 0.2, *p* = 0.004), whereas no trend was evident in the number of seeds (0.24 ± 0.2, *p* = 0.3). Subpopulations differed in their reproductive output accounting for the interannual variation. The two higher subpopulations had a greater production of seed primordia per flower (particularly at S3), and were more similar to each other than to the lowest subpopulation (χ^2^ = 7576.60, df = 3, *p* < 0.001). All subpopulations differed significantly in the number of viable seeds produced per fruit (χ^2^ = 1413.30, df = 3, *p* < 0.001), which increased with elevation. The number of aborts only differed at the extremes of the elevation gradient (χ^2^ = 397.08, df = 3, *p* < 0.001) and the proportion of aborts-primordia was consistently higher in the lower site.

Moreover, the maximum mean values obtained for seed primordia per flower corresponded to site S3 in 2005 (20.1), while in 2020 values found in all three sites were slightly higher than 17.15 (Figure 2a). Regarding the ratio of viable seeds per fruit, the highest mean values were found in S3 in all sampling years followed by site S2, and site S1. It was striking that the mean number of viable seeds appeared to slightly increase over time at site S1, recording its highest value (7.67) in 2020 while it was the opposite for S3, which recorded the lowest mean value (9.96) in 2020 from all the periods sampled (Figure 2b).

On the other hand, the results of the ANOVA analysis showed statistically significant differences in the values of seed primordia per flower between sampling sites S1 and S3 in 2005, 2013, and 2014. However, this condition was not met for the 2020 data (Table 2). Similarly, Table 3 shows that there were significant differences in the data for viable seeds between sites S1 and S3 in the years 2005, 2013, and 2014. These differences were not found for the 2020 data. However, the 2020 data for S3 showed statistically significant differences to those for S1 in 2005, 2013, and 2014.

Regarding reproductive success results, the optimum efficiency in seed production related to the seed primordia was at site S3 for all periods (Table 4). The lowest reproductive success corresponded to site S1 in all the samplings, being the minimum value in the year 2014, with 36.73%, this also occurred for S2 and S3, whose minimums were 41.92% and 55.2%, respectively.

### 2.2. Current SDMs for M. fontqueri

The segmented modeling of occurrences in the three elevational ranges corresponding to sites S1, S2, and S3 served for fine-tuning the selection of predictor variables. On the other hand, the total distribution area of the species was modeled with all the available occurrences and the most explanatory environmental variables.

The model obtained for the subpopulation at a lower altitude (Figure 3A) got an Area Under the Curve (AUC) value of 0.972. Thirteen occurrences of the species recorded in this altitudinal range were used. The variable that relatively contributed the most to explaining the model was the Enhanced Vegetation Index (EVI) with 75%, followed by SLOPE (9%) and Normalized Difference Snow Index (NDSI) (7.5%). In the jackknife test, the variables EVI, NDSI, and SLOPE were the most explanatory; however, the regularized training gain values of the environmental variables were, in general, lower than for the other two subpopulations at higher altitudes.

The model for occurrences in the intermediate altitudinal band (Figure 3B) had an AUC of 0.989. In this case, 99 occurrence records were used. The variable that relatively contributed the most to the model was Average Summer Precipitation (ASP) (38%), followed by Average Annual Temperature (AAT) (15%), EVI (13%), and NDSI (8%). In the jackknife test, the variables that explained the most in isolation were ASP, followed by AAP (Average Annual Precipitation), AAT (Average Annual Temperature), and AST (Average Summer Temperature) with slightly lower values, and the altitude (Digital Elevation Model: DEM).

The model for the upper altitudinal range was run with 66 occurrences of the species (Figure 3C) and yielded an AUC of 0.991. The relatively highest contributing variable was ASP (59%). In the jackknife test the variable that isolated and explained most of the model gain was ASP followed by altitude (DEM) and the climatic variables AAP, AST, AAT, and Potential Evapotranspiration (PET).

The model for the current potential distribution range of the species (i.e., combining the three subpopulations) obtained an AUC of 0.975. The relatively highest-contributing variable to the model was ASP with 39%, followed by NDVI (15%) and NDSI (8.4%). The jackknife test result showed NDSI was the variable with the highest self-explanatory gain, followed by EVI and ASP, closely followed by winter temperature (Figure 4).

Table 5 shows the results for the area (km^2^) occupied by the pixels of maximum suitability for the models run according to elevational ranges and for the total distribution area. The largest area of maximum suitability was obtained from the model for the total occurrence (79.98 km^2^). Counterintuitively, when comparing the three elevational ranges, the highest habitat suitability and the largest area were found in the model for the lowest altitude belt (0.30 and 44.30 km^2^, respectively).

### 2.3. SDMs for M. fontqueri in Future Scenarios

Figure 5 shows the modeling results of the potential habitat distribution of *M. fontqueri* considering increases in both temperature and precipitation (RCP 2.6, RCP 8.5) for the years 2035 and 2100. All AUC values of the future models were above 0.987 (Table 6).

Table 6 shows how in the most optimistic scenario the potentially suitable habitat could be reduced to 49 km^2^, with 28% of the occurrences excluded from this area. This reduction increased as more pessimistic and longer-term scenarios were analyzed, up to the period 2100 when habitat could decrease to 18.09 km^2^ (18% of current potential habitat), which would result in 81% of occurrences being excluded from the area of maximum suitability. The minimum altitude at which the subpopulations are found in the case of the polygon corresponding to the current model was 1637 m asl while its value increases to 2051 m asl for the most optimistic scenario in 2035, rising to 2259 m asl in the most pessimistic scenario modeled for the year 2100.

## 3. Discussion

Climate change is altering the global biota, potentially leading to individual adaptation and natural selection to the new environmental conditions (metabolism, phenology, etc.), or causing local extinctions and migration altering species’ spatial distribution [8,10,11,12]. Our study contributes to understanding how species with narrow ecological requirements and restricted distribution ranges (i.e., a local endemic of a high mountain island-habitat) are altered in their reproductive fitness and spatial distribution range by changes in environmental conditions. In addition, as Lee-Yaw et al. [48] proposed, demographic studies are performed to complement SDMs when the goal is to identify populations with the best chance of long-term persistence.

### 3.1. Reproductive Fitness of M. fontqueri

Analyses of reproductive fitness showed the biological optimum for *M. fontqueri* occurs in its upper distribution range, suggesting that an expected rise in temperatures is likely to compromise the viability of subpopulations at lower elevations. The studied subpopulation always exhibited the worse performance at the lowest altitude for all reproductive variables, in line with other studies [49,50], suggesting a higher sensitivity to year-to-year fluctuations of the climatic conditions. This has been observed previously in the seedling recruitment of other Mediterranean mountain plants [50].

Although ANOVA showed significant reproductive differences at the altitudinal extremes of the population, a slight increase in S1 (lowest altitude) values was observed from 2005 to 2020. This suggests a potential convergence in the reproductive fitness between the extremes of the population.

This closer reproductive fitness could be due to an acceleration of phenology at lower altitudes. This phenological disparity could increase over time if environmental effects were more intense in subpopulations at lower altitudes that would advance their phenology. This pattern has been reported in other mountains [51,52] generally affecting the species for which climatic conditions differ strongly across populations. It is worth noting that accelerated phenology (i.e., flowering earlier) does not necessarily improve reproductive success [49] as reproductive structures can be exposed to inadequate conditions (e.g., drastic temperature drops following early reproductive onset or sustained high temperatures for optimal maturation) [51] affecting all subpopulations across their altitudinal range [53].

However, the seed-set results for S3 showed the highest mean production in all cases. The slight decrease in the production of reproductive structures on average in S3 appeared not to represent a decrease in reproductive success compared to S1 and S2. Thus, although fewer primordia and viable seeds might be produced on average at site S3, the produced primordia eventually reach maturity in a higher proportion than at the other sites sampled. This confirms that S3 is the optimal site for the reproductive effectiveness of the species.

The study on reproductive success would require further monitoring to confirm the observed trends, as well as to extend the sampling to several dates to cover the phenological optima of the different locations. In the current context, it is important to pay attention to the reproductive trends (e.g., advanced flowering and fruit set) of subpopulations, especially in the face of climate change effects [31].

### 3.2. Current and Future Habitat Suitability

Modeling the ecological niche of flora species with such a narrow habitat specificity is particularly challenging, as variations in microenvironments can strongly influence habitat suitability [54,55]. In this sense, obtaining detailed information about physiological and ecological factors affecting the species can improve the selection of the most relevant predictor variables in a biological sense, as well as the interpretation of the models results. Likewise, a sufficiently representative number of occurrences guarantees the stabilization of the model results [56]. It is therefore essential to have a broad knowledge of the species, which in this case comes from fieldwork carried out over more than 30 years [57].

A key issue for modeling is to know the ecology of the species [58] in order to choose the environmental variables that best explain its occurrence [35,59]. Another essential issue is to handle such information at the optimal resolution when forecasting habitat suitability at the local scale, especially if the species requires very specific ecological conditions and is restricted to narrow areas [23,54,60]. Due to its characteristics, the ecological niche of *M. fontqueri* can be considered extremely sensitive to spatial changes in the factors controlling its distribution [58,61]. Here, we used the highest pixel resolution available [35,54,58] and detailed information on the actual distribution range of the species, both necessary to model on a local scale the habitat suitability of rare species with very narrow ecological requirements [54,62,63].

The model results by elevation showed a large area meeting suitable environmental conditions at 1600–1900 m asl. This result can be explained simply because the available area at lower elevations is larger than at upper elevations. Additionally, this lowest-elevation model (Figure 3A) excluded the highest elevations as suitable habitats where the species has been confirmed, probably due to the small number of occurrences used and contrasting environmental conditions at the extremes of the species’ range [9,64]. In turn, model results for higher altitudes suggested that the effect of other topographic or vegetation-related variables could be diluted with altitude, and that climate variables combined with altitude itself became more important. This was the case for summer precipitation (ASP), which is considered a limiting factor, especially in Mediterranean regions [20,33,65]. The variable NDSI, which depends on the snowfall period, the amount of snowfall, and the time of melting [51,66,67], appeared among the four that contributed most to the explanation for the models and could be a relevant predictor of the behavior and potential distribution of the species. The variable SLOPE was also highly explanatory, consistent with the habitat preference of *M. fontqueri* for vertical cliff walls and overhangs, mainly on northern exposures.

The results of the model calculated with all occurrences largely overlapped with the sum of the models for each altitudinal range; however, a larger area with optimal habitat suitability was observed. Moreover, it was consistent with the information obtained in the reproductive biology results. The optimum for potential habitat suitability was found in the higher distribution margins, consistently with the maximum reproductive effectiveness (S3). The easternmost area and northern exposure were confirmed as the most suitable for potential habitat. The minimum altitude of the potential habitat was restricted to approximately 1600–1900 m asl, confirming that its optimum is above this range. The most important variable contributing to the model was summer precipitation (ASP), in addition to EVI and NDSI. EVI was the most relevant variable for the lower altitudinal range and the second for the intermediate range, contributing also to the total model, together with NDSI, which appeared as the third most relevant variable. This result is congruent with previous studies and again highlights the importance of the presence of snow and summer precipitation for the presence of *M. fontqueri*. Engler et al. [16] showed that projected habitat loss is higher for species distributed in high mountains; in this case study, it was scenario-dependent, namely, 80% habitat loss for 36–55% of alpine species between 2070 and 2100. The results of the modeling of the future habitat for *M. fontqueri* concurred with these results, showing that the maximum potential suitable habitat could be reduced by up to 80% by 2100. The minimum altitude where the presence was estimated for the different scenarios indicates the loss of suitable conditions at lower altitudes and the contraction of the habitat. These results agree with several studies warning about the possible local extinction of populations due to the contraction of the lower limit of their habitat [1,26,45,68,69,70], partly explained in the context of climate change by the movement of species towards optimal conditions [71].

The contraction of species ranges in the current climate scenario is mainly owed to the variation in precipitation and temperatures [68]. Particularly, in the Mediterranean region, summer temperatures and mean annual precipitation appear as the most relevant variables that, in the future, could determine the ability of species to withstand environmental conditions, especially at their lower limits of distribution [23,26].

It has been found that changes in environmental variables can affect the growth of populations [72]. Temperature, radiation, photoperiod, or timing of snowmelt are important signals to optimize flowering, especially in alpine plants [66]. Several studies suggest that some plant species are experiencing an acceleration in their phenological processes, which would mean that they have less time after snowmelt to acquire resources before environmental signals trigger the onset of flowering [49,51,67]. In addition, under conditions of reduced water availability, flower longevity tends to decrease to optimize water loss [73], reducing reproductive effectiveness. In the studied case of *M. fontqueri* the environmental variables NDSI (snow index), AST (summer temperature), AWT (winter temperature), and AAP (annual precipitation) contributed significantly to the results of the SDMs and could, therefore, be assumed to be decisive factors for both the distribution of potential habitat and the reproductive mechanism of the species. Nonetheless, the uncertainty of remote sensing data, including EVI and LST [74,75,76], may affect the research results.

## 4. Materials and Methods

### 4.1. Study Area

The study area is located in the eastern part of Sierra Nevada (SE Spain; latitude 37.0275° N; longitude −2.9232° W) (Figure 6). Sierra Nevada has a complex orography and soil composition distributed along 2100 km^2^, comprising altitudinal ranges from 200 to 3482 m asl. The climate is Mediterranean, characterized by cold and wet winters and hot and dry summers. The average annual rainfall is highly variable ranging from 300 to 1000 mm, with a high spatial variability due to topographic effects [77]. Average temperatures are below 0 °C during winter with a snow cover that can persist up to 8 months in the highest areas (occasionally up to 10 months in small snow patches) [24]. The region has experienced an increase in temperatures and a greater variation in rainfall during the last decades [77].

### 4.2. Study Species

*Moehringia fontqueri* is an endangered caespitose hemycriptophyte strictly endemic to the Eastern part of Sierra Nevada (Almeria) occupying a range from 1600–2430 m asl (data in this study) in the oromediterranean bioclimatic belt under subhumid ombroclimate (precipitation between 600–1000 mm). It grows in crevices of north-faced cliffs under no direct sunlight, being part of chasmophytic vegetation (*Centrantho nevadensis-Sedetum brevifolii* in Quézel 1953). This species germinates and sprouts in mid-May, and blooms from early June (at lower altitudes) to early August (at higher altitudes), with flowering peaking around mid-July. Most individuals fruit in August and seeds are dispersed thereafter [61]. This species has one single known population (115,000 estimated individuals) fragmented into approximately 70 subpopulations with a possible genetic exchange. Its geographic range is 200 km^2^, and its estimated area of occupancy (AOO) is between 2.4 and 10 km^2^. The species has been categorized as Endangered (En B1ac (iv) + 2ac (iv)) [47].

### 4.3. Reproductive Biology

Field surveys were carried out on foot by expert staff who regularly visited the different subpopulations. In this work, 178 presence records were located in the study area, distributed in three main elevations, with 13 records between 1600–1900 m asl (S1), 99 between 1900–2200 m asl (S2), and 66 between 2200–2500 m asl (S3) (Figure 6). Field surveys for reproductive data were distributed in three main elevational ranges, 1960 m asl (S1), 2150 m asl (S2), and 2430 m asl (S3). On each elevational range, in summers of 2013, 2014, and 2020, at least 30 flowers and 26 fruits were collected from several individuals and preserved in the field in Kew Mix (53% methanol, 37% deionized water, 5% formaldehyde solution, and 5% glycerol), fruits were harvested individually in cellophane bags, to allow transpiration and dry (preventing them from rotting). Flowers and fruits were dissected individually using a Motic binocular stereo-microscope (10×–30×). Reproductive structures were counted evaluating the numbers of seed primordia per flower and viable seeds per fruit (Figure 7 and Table 7).

Two reproductive variables were calculated considering each location by the altitudinal range and sampling year to assess reproductive success [78]: Method calculations of the seed-set as in Equation (1), and abortion rate as in Equation (2) (where ‘Sx’ is the site under study (x = 1, 2, 3), and ‘a’ the year (a = 2005, 2013, 2014, 2020)).
[[∑ Viable seeds in S (x,a)/N Fruits in S(x,a)]/[∑ Seed primordia in S(x,a)/N Flowers in S(x,a)]] × 100(1)
[[∑ Aborted seeds in S (x,a)/N Fruits in S(x,a)]/[∑ Seed primordia in S(x,a)/N Flowers in S(x,a)]] × 100(2)

### 4.4. Data Analyses

Temporal trends in the reproductive variables across sites were evaluated by fitting General Linear Models (GLMs) including the effects of the site, year, and their interaction as explanatory variables. Linear models were built with a Gaussian family error distribution using the R package ‘stats’ [79]. Significant interactions were compared with Tukey’s post hoc analysis using the R package ‘agricolae’ [80]. Anova analyses helped to further identify reproductive differences between particular sites and years. Graphs were produced using ‘ggplot2′ in the R package ‘tidyverse’ [81].

We tested the general trends over time for the reproductive variables of *M. fontqueri* considering all subpopulations together. Alternatively, we determined whether there were reproductive differences between subpopulations accounting for the variation between years. Thus, we fitted Linear Mixed Models (LMMs) considering the year as a fixed factor and site as a random effect, and conversely, with the site as a fixed factor and year as a random effect. The use of LMMs allowed us to determine the effect of year and site controlling, respectively for spatial or temporal pseudo-replication. All mixed-effects models were built with a Gaussian error distribution, and identity-link function using the R package ‘glmmTMB’ [82]. Multiple comparisons were performed with the R package ‘multcomp’ [83].

For all models, time was used as a continuous variable. Appropriate distribution for the reproductive variables was determined by inspecting the histograms of the data values with the R package ‘graphics’ and likelihood ratio tests between models using different distribution families with the R package ‘stats’ [80]. We used the R package ‘DHARMa’ [84] to test for uniformity, outliers, and overdispersion on the scaled residuals and validate the goodness of the models. All statistical analyses were performed using R version 4.1.0, R Foundation for Statistical Computing (Vienna, Austria) [79].

### 4.5. Current and Future Predictor Variables

This process was used to try to discriminate potentially suitable sites for this ecological requirement of the species. Environmental variables (30 m^2^ resolution) were generated on the Google Earth Engine platform (https://earthengine.google.com (accessed on 21 July 2021)). The variables Aspect (ASP), Hillshade (HSD), and Hillshadow (HSW) were obtained from a Digital Elevation Model (DEM) [85,86]. Considering the ecology of the species (which occupies steep microsites), the Slope variable (SLP) was generated in QGIS from the DEM at 5 m^2^ resolution (available at https://www.ign.es/web/seccion-elevaciones (accessed on 10 November 2020)). The plugin ‘ZonalStatistics’ was used to obtain the maximum slope values in the 5 m^2^ pixel size raster. A downscaling process was performed by the ‘Polygon To Raster’ tool to obtain a 30 m^2^ pixel scale grid, to which the maximum SLP value for all the 5 m^2^ pixels contained in each 30 m^2^ grid was assigned.

On the other hand, satellite variables were preprocessed and generated on Google Earth Engine from Landsat 8 satellite image collections (USGS Landsat 8 Level 2, Collection 2, Tier 1, 2013 to 2021). Enhanced Vegetation Index (EVI) layers [87] were produced using the 8-day Landsat8 images, while Normalized Difference Snow Index (NDSI) for the winter period [88], Normalized Difference Water Index (NDWI) for the summer period [89], and Land Surface Temperature (LST) [90] were calculated from annual Landsat8 images. For these indices, interannual mean, median, and SD values for the pixel information were calculated independently.

The scarcity of meteorological information at the chosen resolution, which is common in high mountain areas [64], led to the production of climate layers, as they are considered essential for forecasting species behavior [39,91]. Meteorological data from 17 virtual stations located in the study area were downloaded from the website http://diagramasbioclimaticos.com (accessed on 17 April 2021). Temperature, precipitation, and potential evapotranspiration data were interpolated on QGIS by ‘krigging regression’ plugin with height (DEM altitude) as the covariate. For the estimation of future conditions, the different raster layers of climate variables were modified by adjusting the values to the change predicted by the global change scenarios [37], RCP 2.6 and RCP 8.5 in 2035 and 2100 (Table 8).

To avoid collinearity issues in the sets of predictor variables, and in order to select only non-redundant ones, a Variance Inflation Factors (VIF) analysis was performed on R software [92,93,94,95,96,97,98]. According to this methodology, the value for the VIF analysis must be less than 5 (or additional variables with a high VIF may be excluded until all remaining variables had VIF < 10), and in agreement with Dormann et al. [99], a correlation below 70 in the environmental variables can be considered as an acceptable value (see Appendix A). This analysis was used to discern between predictor variables in modeling processes both for current and future conditions.

For the current models, predictor variables SLP, EVI, HSD, DEM, NDSI, THL, and NDVI were selected. Due to the high correlation of climate variables, an isolated VIF analysis was required to discriminate between them. PET (Potential Evapotranspiration), AAP (Average Annual Precipitation), AWP (Average Winter Precipitation), ASP (Average Summer Precipitation), AAT (Average Annual Temperature), AWT (Average Winter Temperature), and AST (Average Summer Temperature) were selected as well.

### 4.6. SDMs and Ecological Niche

We modeled habitat suitability for *M. fontqueri* using MaxEnt (based on the maximum entropy principle) version 3.4.1k, American Museum of Natural History, Center for Biodiversity and Conservation (New York, EEUU) [100]. This software generates empirical models based on statistical or theoretical response surfaces [32], also defined as statistical models that use observed distribution data to infer ecological requirements and map their current potential distribution [101], and project into the future under conditions of climate change [102]. The probability of suitability for each pixel given a sample of the background is calculated following the idea that the expected value for each variable must be equal to the empirical average value of current occurrences of the species [100,102,103]. We used MaxEnt given its suitability for presence-only data (see Elith et al. [102]). The default values were kept by default, following the methodology of most related publications [104], except that the value of 1000 iterations were entered, and the program was required to output response curves for the variables, assessed with a jackknife test [105]. Initially, we ran models using three groups of presence records corresponding to the altitudinal ranges (S1, S2, S3) for model calibration using the whole set of environmental variables (see above). This step allowed us to evaluate a diverse array of candidate models, selecting the best models. To estimate the predictive capacity of each model and the effect of each predictor variable, the area under the curve (AUC), which indicates how far the result departs from a null model, was evaluated following the scale recommended by Swets [106]: AUC < 0.5 low accuracy; 0.5 probability with no effect of the variables; 0.5–0.6 not predictive; 0.6–0.7 poor accuracy; 0.7–0.8 fair accuracy; 0.8–0.9 good accuracy; >0.9 excellent accuracy. The fact that the AUC decreases when a predictor variable is excluded might suggest that the other predictors are not able to explain the projected distribution as accurately as if it is retained. Conversely, it suggests that its inclusion has caused the model to over-fit the field data, and therefore, explains no more than the other factors do [107].

The Logistic output format was chosen (habitat suitability between 0–1 for each pixel). Both plots of predictor variables gains and jackknife tests determined how much each variable could explain changes in model outcomes. A table with percentages of the relative importance of each variable was also produced. To compare the changes between the different models, shape layers with vector information were created in QGIS [108]. Polygons were generated using the criterion of including only pixels with values in the first four quartiles of potential habitat suitability based on the model results for *M. fontqueri*. The area (km^2^), mean polygon suitability, SD, and variance were calculated for the different polygons of the highest suitability, so that an absolute comparison could be made between them.

## 5. Conclusions

Multidisciplinary studies considering reproductive biology and the use of SDMs under different climate change scenarios are key tools to understand the status and performance over time of plant species populations in the Mediterranean highlands [31,69]. These studies are able to generate relevant information for the design of conservation plans adapted to the context of global change [1,20,32,36,101,109] prioritizing especially endemic orophilous taxa [27,54].

*M. fontqueri* subpopulations at the lowest altitudinal ranges appeared to experience a decrease in their reproductive success while the highest subpopulations seemed to reproduce effectively. Although the decrease in reproductive success in the long term cannot be confirmed, our results showed that *M. fontqueri* strongly depends on the environmental conditions, suggesting that the alteration of climate under future scenarios poses a risk to its conservation, as forecasted by the species distribution models.

Regarding the models generated for current conditions, the amount of snowfall was an explanatory variable present in all models. In addition, the variables ASP (summer precipitation), NDSI (amount of snowfall), and EVI (vegetation index) contributed more significantly than others to the distribution model of *M. fontqueri*. According to current models, the highest values of potential habitat suitability would be found at higher altitudes, while habitat suitability at lower altitudes is expected to decrease. Furthermore, according to these models, the predictor variables explaining future habitat suitability are mean summer temperature, mean winter temperature and mean annual precipitation.

The projections showed a drastic decline in the distribution range of the species that would mainly affect subpopulations at lower elevations. This is consistent with studies indicating that peripheral populations of a species or in their distribution limit are more sensitive to any alteration since they meet their optimum ecological requirements with more difficulty [110,111]. This could cause the limit of the distribution to shift upwards in the future, as widely reported for other species [10,11], increasing the probability of extinction by depletion of suitable habitat.

The analysis of the predictions of changes in climatic conditions, and the coincidence of the area of greatest reproductive success with the areas where, according to the models, the species will have ideal conditions for the longest time suggests that the high areas could serve as the main refuge for the species and a reservoir of a greater gene pool in the future. In turn, at lower altitudes, the persistence of the species will be compromised due to declining habitat suitability resulting from the decreasing reproductive success and altered phenology of the species. However, long-term monitoring would be necessary to assess these trends in the future.

Threatened species and rich biodiversity areas have been protected by law in the past, although this protection has failed to consider climate as a modeler of biodiversity over time. In this sense, protection planning should consider the potential future constraints, implying not just current but future patterns of biodiversity [112].

Our simultaneous assessment of reproductive success and habitat suitability aims to serve as a model to guide conservation, management and climate change mitigation strategies through adaptive management to safeguard the persistence of the maximum genetic pool of Mediterranean high-mountain plants threatened by climate change. For this reason, we emphasize the need to devote effort to the conservation of other mountain species with a perspective for the future, that as *M. fontqueri*, may see their status worsened by potential environmental changes due to climate change.

## Figures and Tables

**Figure 1 plants-11-03193-f001:**
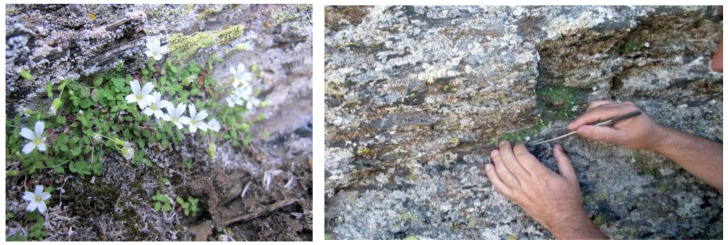
*Moehringia fontqueri*, plant detail (**left**) and sample collection method (**right**).

**Figure 2 plants-11-03193-f002:**
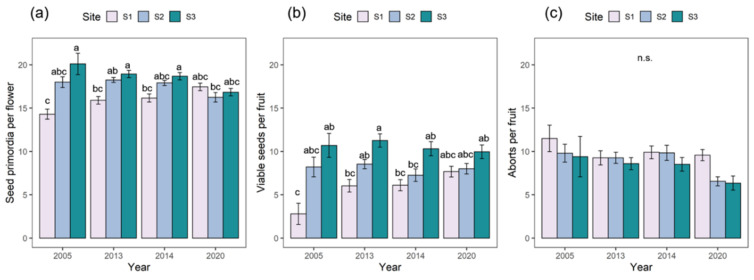
(**a**) Means of seed primordia per flower, (**b**) viable seeds per fruit, and (**c**) aborts per fruit by studied site over time (±SE). Different letters represent significant differences between sites at *p* < 0.01 in the post hoc Tukey tests after GLMs.

**Figure 3 plants-11-03193-f003:**
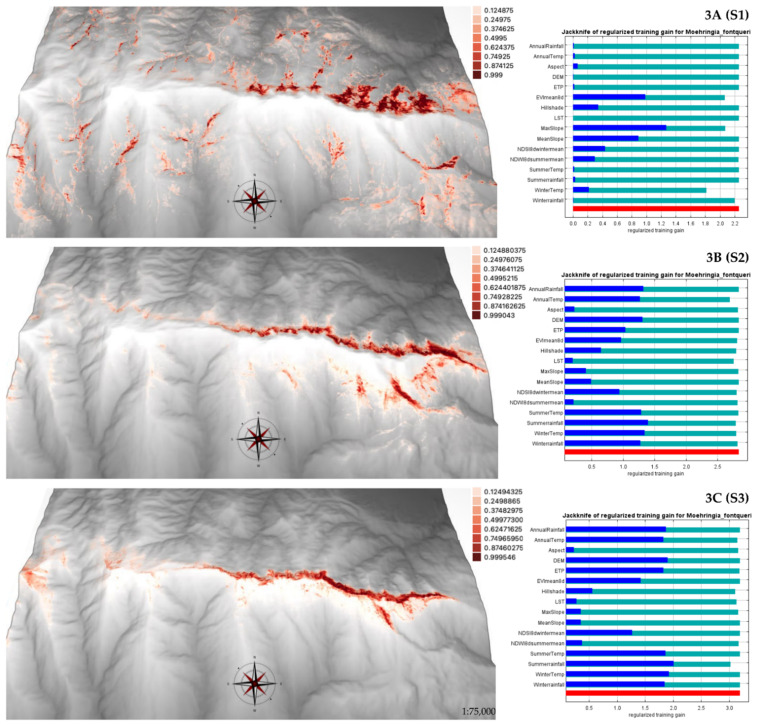
Models of current potential distribution of *Moehringia fontqueri* derived from occurrences segregated by elevational ranges **3A (S1)**, **3B (S2)**, **3C (S3)**. Regularized training gain values of the jackknife tests.

**Figure 4 plants-11-03193-f004:**
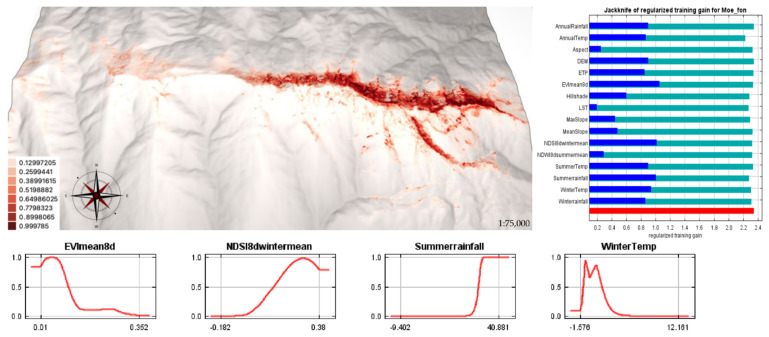
Current potential distribution model of *Moehringia fontqueri* for all occurrences throughout the distribution range. Regularized training gain values of the jackknife tests. Most explanatory variables (ASP: Summerrainfall (mm) AWT: WinterTemp (°C)).

**Figure 5 plants-11-03193-f005:**
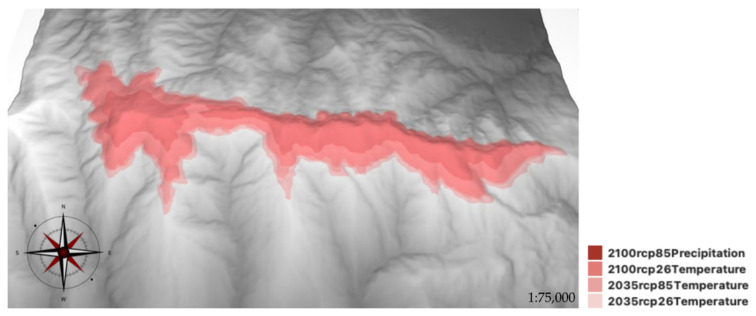
Reduction estimation of the maximum potential habitat suitability for *Moehringia fontqueri*. Temperature-based RCP 2.6 and precipitation-based RCP 8.5 scenarios for the years 2035 and 2100.

**Figure 6 plants-11-03193-f006:**
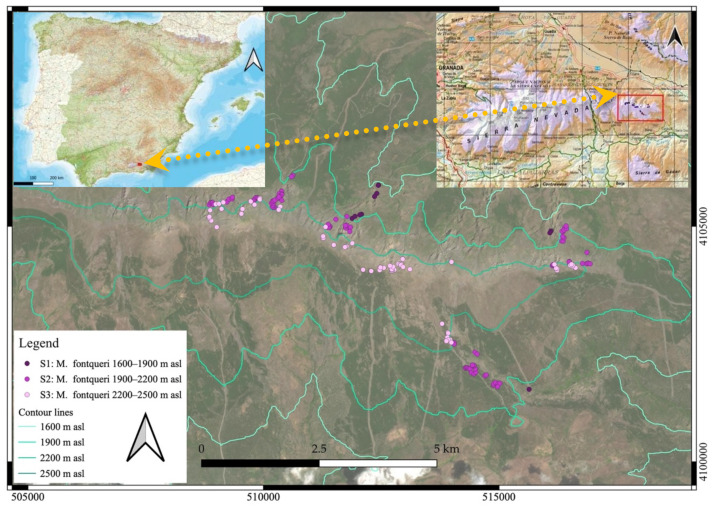
Distribution range map of *Moehringia fontqueri*. Detail of the distribution of occurrences in Sierra Nevada divided by altitude (S1, S2 and S3).

**Figure 7 plants-11-03193-f007:**
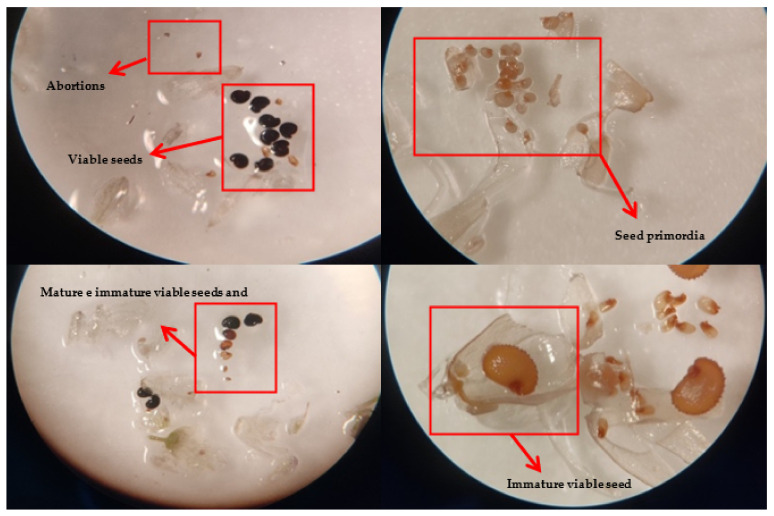
Images of the dissection process of flowers and fruits during the study of seed primordia and mature seeds.

**Table 1 plants-11-03193-t001:** Generalized linear models testing the effects of site, year, and their interaction on the number of seed primordia per flower, number of viable seeds per fruit, and number of aborted seeds per fruit. Results with *p* < 0.05 in bold.

		Seed Primordia per Flower		Viable Seeds per Flower		Aborted Seeds per Fruit	
	df	χ^2^	*p*	χ^2^	*p*	χ^2^	*p*
Site	2	23.11	**<0.001**	63.69	**<0.001**	9.73	**0.01**
Year	1	4.05	**0.04**	2.69	0.10	11.31	**<0.001**
Site × Year	2	38.74	**<0.001**	11.06	**<0.001**	1.91	0.39

**Table 2 plants-11-03193-t002:** Statistically significant differences from ANOVA analysis for seed primordia data. *p* < 0.05.

Hypothesis	95% Confidence Intervals for Group Differences	*p*-Value
S1 2005–S2 2013	−7.2610–−0.6388	0.0036
S1 2005–S3 2013	−7.9930–−1.2820	<0.001
S1 2005–S2 2014	−6.9620–−0.2504	0.0156
S1 2005–S3 2014	−7.7090–−1.0440	<0.001
S1 2005–S1 2020	−6.3550–0.0526	0.0428
S1 2005–S3 2005	1.6570–9.9430	<0.001
S3 2005–S1 2013	0.8281–7.5650	0.0017
S3 2005–S1 2014	0.5510–7.3160	0.0053
S1 2013–S2 2013	−4.6160–−0.0771	0.0248
S1 2013–S3 2013	−5.3690–−0.6999	<0.001
S1 2013–S3 2014	−5.0740–−0.4729	0.0030
S2 2013–S2 2020	−0.0423–4.0420	0.0447
S3 2013–S1 2014	0.4168–5.1250	0.0044
S3 2013–S2 2020	0.5735–4.8010	0.0012
S3 2013–S3 2020	−0.0489–4.2350	0.0457
S1 2014–S3 2014	−4.8300–−0.1895	0.0143
S3 2014–S2 2020	0.3501–4.5030	0.0049

**Table 3 plants-11-03193-t003:** Statistically significant differences from ANOVA analysis of viable seeds data. *p* < 0.05.

Hypothesis	95% Confidence Intervals for Group Differences	*p*-Value
S1 2005–S3 2005	2.1290–13.6700	<0.001
S3 2005–S1 2013	−0.0250–9.3610	0.0379
S3 2005–S1 2014	−0.1120–9.3120	0.0459
S1 2005–S1 2020	−9.4800–−0.2536	0.0196
S1 2005–S2 2013	−10.4500–−1.0210	0.0026
S1 2005–S2 2020	−9.9770–−0.4227	0.0133
S1 2005–S3 2013	−13.2400–−3.6820	<0.001
S1 2005–S3 2014	−12.2400–−2.7780	<0.001
S1 2005–S3 2020	−11.9600–−2.3590	<0.001
S1 2013–S3 2013	−8.6240–−1.8300	<0.001
S1 2013–S3 2014	−7.6120–−0.9441	0.0010
S1 2013–S3 2020	−7.3610–−0.4974	0.0066
S1 2014–S3 2013	−8.5830–−1.7360	<0.001
S1 2014–S3 2014	−7.5710–−0.8496	0.0016
S1 2014–S3 2020	−7.3190–−0.4036	0.0094
S1 2020–S3 2013	−6.8780–−0.3071	0.0125
S2 2014–S3 2013	−7.3980–−0.6040	0.0044

**Table 4 plants-11-03193-t004:** Percentage of reproductive effectiveness (seed-set) for each year at the three sites sampled.

Seed-Set	2005	2013	2014	2020
S1	20.84	37.38	36.73	44.00
S2	45.32	49.55	41.92	49.23
S3	57.10	59.45	55.20	59.14

**Table 5 plants-11-03193-t005:** Results for area (km^2^) and mean suitability of potential habitat included in the polygon of the current models.

Elevational ranges	Area (km^2^)	Average Potential Habitat Suitability	SD	Variance
S1 (1600–1900 m asl)	44.30	0.30	0.24	0.05
S2 (1900–2200 m asl)	36.58	0.16	0.18	0.03
S3 (2200–2500 m asl)	34.41	0.16	0.17	0.03
Total study area	79.98	0.15	0.18	0.03

**Table 6 plants-11-03193-t006:** Area (km^2^) occupied by the polygons of maximum current and future habitat suitability. Occurrences that would be found within each delimited area. Minimum altitude at which suitable habitat would be found for the species. Average suitability of the habitat in the subsequent polygons for each model (±standard deviation). AUC of each model.

Scenarios	Area (km^2^)	Occurrences in Maximum Suitability Areas	Minimum Altitude (m asl)	Average Suitability ± SD	AUC
Current	96.75	177	1637	0.1330 ± 0.0200	0.975
2035 RCP 2.6	49.00	128 (−49)	2051	0.0039 ± 0.0004	0.988
2035 RCP 8.5	41.6	110 (−67)	2113	0.0035 ± 0.0004	0.987
2100 RCP 2.6	37.47	103 (−74)	2113	0.0029 ± 0.0005	0.987
2100 RCP 8.5	18.09	35 (−142)	2259	0.0140 ± 0.0123	0.991

**Table 7 plants-11-03193-t007:** Number of flowers and fruits collected at each elevational range per year.

		Number of Flowers				Number of Fruits			
Main Locations Name and Elevation (m asl)	Elevational Ranges	2005	2013	2014	2020	2005	2013	2014	2020
S1: Barranco de la Campana (1960)	S1: 1600–1900	10	31	30	52	10	36	32	36
S2: Between La Polarda and El Buitre (2150)	S2: 1900–2200	10	36	32	48	10	30	31	27
S3: Under El Buitre summit (2430)	S3: 2200–2500	10	32	34	45	10	27	29	26

**Table 8 plants-11-03193-t008:** Calculation of temperature and precipitation variation for each Global Change scenario studied.

Scenario	Average Annual Temperature	Average Winter Temperature	Average Summer Temperature	Average Annual Precipitation	Average Winter Temperature
2035 RCP2.6	+0.98	+0.80	+1.32	−1.40%	−1.50%
2035 RCP8.5	+1.16	+0.86	+1.48	−1.20%	−1.80%
2100 RCP2.6	+1.18	+0.82	+1.58	−3.00%	−2.80%
2100 RCP8.5	+4.88	+3.96	+6.18	−18.40%	−14.40%

## Data Availability

Not applicable.

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
