# Peer review of "The Fate of Endemic Species Specialized in Island Habitat under Climate Change in a Mediterranean High Mountain"

_plants, 2022, doi:10.3390/plants11233193_

Round 1
Reviewer 1 Report
Dear Authors,
To my view, your manuscript on the reproductive success and habitat suitability of Moehringia fontqueri under climate change scenarios includes significant and worth published data, properly analysed and discussed. Your results on the reproductive biology of the species are effectively combined with the habitat suitability analyses, emphasizing the challenges that Mediterranean mountain endemics face because of climate change. I also really appreciate the field work you have contacted collecting all these valuable data.
Thus, I consider your manuscript suitable for publication in Plants after minor revisions. All my suggestions for the improvement of your manuscript are included in the attached file.
Sincerely

Author Response
We would like to thank the reviewers for all their suggestions and valuable comments on the revision of this article. We would also like to thank the editor of the journal Plants for his work in handling the manuscript and expediting the process.
The details about the revisions are as follows:
Response to Reviewer 1 Comments
Responses to the reviewer's comments are made on the attached pdf.

Reviewer 2 Report
Based on the data of species occurrences, topographic, satellite and climatic variables, this study analyzed the current habitat suitability of species at different altitudes in the Sierra Nevada, and explored the future habitat suitability of two climate change scenarios (RCP 2.6 and 8.5). The results of this study may contribute to provide some theoretical support for the protection of Mediterranean high-mountain plants threatened by climate change. However, there are some concerns that the authors should address before it can be considered for publication.
(1) The authors did not fully introduce the data and methods. For example, why the authors chose 2005, 2013, 2014 and 2020 as this study period?
(2) L485-487. Why the authors only choose the winter and summer climate variables (Average Winter Temperature and Average Summer Temperature)?
(3) For the satellite variables data, have the authors made quality control or homogenization procedure?
(4) More mechanism explanations are suggested to further explain the effects of different meteorological factors on habitat suitability in the study area.
(5) A paragraph of limitation discussion should be added to clarify the limitation or uncertainty of data and methods in this current study. For example, the uncertainty of remote sensing data including vegetation Index and land surface temperature data (Hussain et al., 2022; Shen et al., 2020, 2022) may affect the research results.
(6) The units for variable should be added in the Figure 4 and 8.
References:
Spatiotemporal variation in land use land cover in the response to local climate change using multispectral remote sensing data. Land, 2022, 11, 595.
Marshland loss warms local land surface temperature in China. Geophysical Research Letters, 2020, 47, e2020GL087648.
Asymmetric impacts of diurnal warming on vegetation carbon sequestration of marshes in the Qinghai Tibet Plateau. Global Biogeochemical Cycles, 2022, 36, e2022GB007396.
Author Response
We would like to thank the reviewers for all their suggestions and valuable comments on the revision of this article. We would also like to thank the editor of the journal Plants for his work in handling the manuscript and expediting the process.
The details about the revisions are as follows:
Response to Reviewer 2 Comments
Based on the data of species occurrences, topographic, satellite and climatic variables, this study analyzed the current habitat suitability of species at different altitudes in the Sierra Nevada, and explored the future habitat suitability of two climate change scenarios (RCP 2.6 and 8.5). The results of this study may contribute to provide some theoretical support for the protection of Mediterranean high-mountain plants threatened by climate change. However, there are some concerns that the authors should address before it can be considered for publication.
(1) The authors did not fully introduce the data and methods. For example, why the authors chose 2005, 2013, 2014 and 2020 as this study period?
The explanation is related to the fact that the monitoring of this endangered species has been carried out intermittently, as there has not been a single research project covering the whole study period. In addition, due to the pandemic, sampling in 2019 had to be postponed.
(2) L485-487. Why the authors only choose the winter and summer climate variables (Average Winter Temperature and Average Summer Temperature)?
The selection of variables was made on the basis of the results of the jacknife and VIF analyses, taking into account the group of variables that showed a lower correlation index and explained a higher proportion of the resulting model. Appendix 1.
(3) For the satellite variables data, have the authors made quality control or homogenization procedure?
Preprocessing of the satellite image collections was carried out as part of the method of obtaining the indices derived from the satellite variables. The GoogleEngine information repository allows this pre-processing to be carried out before calculating the indices and downloading the raster images that have been loaded into the MaxEnt programme. This idea has been indicated on the manuscript. Line 450.
(4) More mechanism explanations are suggested to further explain the effects of different meteorological factors on habitat suitability in the study area.
Done. We have modified a phrase in the Discussion section. Line 357.
(5) A paragraph of limitation discussion should be added to clarify the limitation or uncertainty of data and methods in this current study. For example, the uncertainty of remote sensing data including vegetation Index and land surface temperature data (Hussain et al., 2022; Shen et al., 2020, 2022) may affect the research results.
We thank the reviewer for his suggestion and have included the paragraph in the Discussion section. Line 359.
(6) The units for variable should be added in the Figure 4 and 8.
Done.
References:
Spatiotemporal variation in land use land cover in the response to local climate change using multispectral remote sensing data. Land, 2022, 11, 595.
Marshland loss warms local land surface temperature in China. Geophysical Research Letters, 2020, 47, e2020GL087648.
Asymmetric impacts of diurnal warming on vegetation carbon sequestration of marshes in the Qinghai Tibet Plateau. Global Biogeochemical Cycles, 2022, 36, e2022GB007396.
Reviewer 3 Report
Dear colleagues!
The authors' research is complex and has fundamental value for understanding the reproduction processes of populations of endemic plant species. The authors carried out a unique, painstaking work on the study of the reproductive biology, phenology and ecological niche of Moehringia fontqueri, modeled the change in the state of the endemic population under various scenarios of climate change in the future.
There is competent and persuasive introduction. The using of modern statistical methods inspires confidence in the results of the research.
The authors also described in detail the results of the work, illustrated their judgments, and conducted a competent discussion of the results.
I have only one comment to the manuscript:
L. 112-113: Why is the arithmetic mean error of the same order as the arithmetic mean itself?
Author Response
We would like to thank the reviewers for all their suggestions and valuable comments on the revision of this article. We would also like to thank the editor of the journal Plants for his work in handling the manuscript and expediting the process.
The details about the revisions are as follows:
Response to Reviewer 3 Comments
Dear colleagues!
The authors' research is complex and has fundamental value for understanding the reproduction processes of populations of endemic plant species. The authors carried out a unique, painstaking work on the study of the reproductive biology, phenology and ecological niche of Moehringia fontqueri, modeled the change in the state of the endemic population under various scenarios of climate change in the future.
There is competent and persuasive introduction. The using of modern statistical methods inspires confidence in the results of the research.
The authors also described in detail the results of the work, illustrated their judgments, and conducted a competent discussion of the results.
I have only one comment to the manuscript:
- 112-113: Why is the arithmetic mean error of the same order as the arithmetic mean itself?
The data resulting from the statistical analysis carried out in the R programme have been taken. By default, the data are shown to two decimal places, but we agree with the reviewer that the order of magnitude of the error may be inappropriate given the values of the variables for the means. The text has been modified to leave only one decimal place.
Thanks again to the reviewers for your valuable comments on this paper.
Round 2
Reviewer 2 Report
The authors have addressed all my concerns. I suggest accept this paper in its present form.